# GATCF: Graph Attention Collaborative Filtering for Reliable Blockchain Services Selection in BaaS

**DOI:** 10.3390/s23156775

**Published:** 2023-07-28

**Authors:** Yuxiang Zeng, Jianlong Xu, Zhuohua Zhang, Caiyi Chen, Qianyu Ling, Jialin Wang

**Affiliations:** 1College of Engineering, Shantou University, Shantou 515063, China; 20yxzeng@stu.edu.cn (Y.Z.); 21zhzhang3@stu.edu.cn (Z.Z.); 21cychen@stu.edu.cn (C.C.); 20jlwang@stu.edu.cn (J.W.); 2College of Science, Shantou University, Shantou 515063, China; 20qyling@stu.edu.cn

**Keywords:** blockchain services, collaborative filtering, graph attention, reliability prediction

## Abstract

Blockchain technology is a decentralized ledger that allows the development of applications without the need for a trusted third party. As service-oriented computing continues to evolve, the concept of Blockchain as a Service (BaaS) has emerged, providing a simplified approach to building blockchain-based applications. The growing demand for blockchain services has resulted in numerous options with overlapping functionalities, making it difficult to select the most reliable ones for users. Choosing the best-trusted blockchain peers is a challenging task due to the sparsity of data caused by the multitude of available options. To address the aforementioned issues, we propose a novel collaborative filtering-based matrix completion model called Graph Attention Collaborative Filtering (GATCF), which leverages both graph attention and collaborative filtering techniques to recover the missing values in the data matrix effectively. By incorporating graph attention into the matrix completion process, GATCF can effectively capture the underlying dependencies and interactions between users or peers, and thus mitigate the data sparsity scenarios. We conduct extensive experiments on a large-scale dataset to assess our performance. Results show that our proposed method achieves higher recovery accuracy.

## 1. Introduction

Blockchain, a decentralized distributed database technology that records and validates transactions and service-oriented computing (SOC) is a software design approach that breaks down applications into reusable services that can be developed, deployed and managed independently. In addition, blockchain may also enable potential improvements in secure embedded systems [1] and dynamic hardware resource allocation [2]. As an integral component of software system development, blockchain technology provides communication, data storage, data mining, and computation services. Microsoft and IBM [3] have introduced uBaaS (Unified Blockchain as a Service) [3], which is based on the concept of service-oriented computing. Blockchain technology has demonstrated its effectiveness in a variety of industries [4], including finance, banking, bitcoin, and healthcare [5]. In the BaaS paradigm, blockchain-based applications can be constructed by invoking numerous blockchain services through the IoT [6]. This approach enables developers to rapidly validate their models and concepts, facilitating the faster development and deployment of blockchain-based applications.

As a benefit, the growth of unified blockchain technology and SOC has led to a proliferation of Blockchain-as-a-Service platforms for domains like the Internet of things, edge computing, and web services. Because the number of services deployed on the blockchain has dramatically increased, it has become confusing for people to select the most suitable blockchain services. Finding the most reliable blockchain service among functionally similar ones poses a special challenge in constructing highly reliable blockchain-based applications. One useful approach to achieve this is to utilize the nonfunctional properties of services to rank and select the most reliable ones. Reliability is considered an essential nonfunctional feature for service selection [7]. These properties are typically used to evaluate and measure the performance and reliability of different blockchain services. However, when a blockchain service recommender system suggests services that a user has not previously used, the reliability values that the user relies on are unknown. In such situations, predicting the unknown reliability of the service is crucial to determine whether the recommended service is appropriate for the user. In real-world scenarios, the number of blockchain peers can exceed 20,000, making it impractical for a single user to establish simultaneous connections with all peers and evaluate their reliability. As a result, users rely on predictive methods to assess the reliability of blockchain peers. According to these problems, how to design a method to obtain the unknown reliability values of candidate blockchain services without invoking them is still an issue.

To address this, some challenges must be taken into consideration:**Higher-order Relationships.** Traditional graph neural networks may have limitations in utilizing collaborative information due to their difficulty in modeling higher-order relationships. These networks typically capture only local n-order neighbor relationships, resulting in a lack of ability to model multi-hop connections or higher-order relationships and a consequent loss of synergy information. This loss may negatively impact the discovery of potential patterns and relationships in graph data, as interactions between nodes often involve multi-hop relationships. Focusing solely on n-order neighbor information may not accurately capture global structures. Therefore, it is crucial to develop a graph extraction architecture that excels at leveraging collaborative information, enabling enhanced extraction of synergistic insights.**High Sparsity.** In high-sparsity scenarios, the performance of both traditional and state-of-the-art neural network models is suboptimal. The presence of sparsely observed data during the initial phase of reliability prediction systems can have significant implications if the accuracy of the predictive model is inadequate. This may lead to adverse outcomes; for example, establishing connections with unreliable blockchain peers can result in resource wastage and even substantial financial losses, potentially amounting to millions of dollars in cryptocurrencies. Therefore, there is an urgent need for an effective reliability prediction model that can operate robustly in highly sparse scenarios.

In the context of blockchain services, collaborative filtering has emerged as a widely adopted approach in QoS-based services recommender systems. This involves learning about user interactions from their historical records. In this paper, we proposed a novel model named Graph Attention Collaborative Filtering (GATCF), which mainly utilizes the graph attention mechanism and collaborative filtering techniques to predict the reliability of user–peer interactions. Our model leverages the graph attention mechanism to effectively capture the dependencies and interactions between user and peer nodes, alleviating the data sparsity problem. The model includes an embedding transfer module and an interaction module for extracting graph structural features and modeling the relationships between potential factor vectors of users and peers, respectively. In order to evaluate the effectiveness of our proposed framework and demonstrate its superiority over existing reliability prediction models, we conducted experiments on a real-world dataset. The main technical contributions are summarized as follows:We propose GATCF, a model that employs a graph attention mechanism to predict the reliability of blockchain peers. Our framework outperforms most existing approaches in terms of prediction performance.We introduce a novel model that combines graph attention mechanisms with collaborative filtering, and investigate the impact of different interaction functions on the overall performance of the model.We extensively evaluate our proposed method on a large-scale real-world dataset, demonstrating the superiority of our proposed work. Specifically, our method outperforms the majority of the existing methods in blockchain services, indicating its promising potential in practical applications.

The remainder of this paper is organized as follows. We review related literature in Section 2. Section 3 introduces the problem formulation and the system model design overview. Section 4 presents our detailed design and complete solution. Section 5 evaluates our model performance under different settings. Section 6 draws the conclusion.

## 2. Related Work

Collaborative filtering (CF) is a widely used technique for predicting the quality of service (QoS) in recommender systems. That can also be used to predict the reliability of peers. However, the method of collecting scattered user raw data and uploading it to the cloud platform for modeling has the risk of leaking privacy. Many people have been working on improving the reliability of the blockchain over the years. Liang et al. [8] introduced a blockchain system for safeguarding circuit copyright using homomorphic encryption. The system ensures secure and scalable circuit copyright protection by ensuring the accurate execution of smart contracts within the blockchain. Lei et al. [9] proposed a reputation-based Byzantine fault tolerance (RBFT) algorithm that employs a reputation model to assess the behavior of each peer during a consensus process. The approach incorporates sparse network monitoring to conduct performance profiling for large-scale systems. Previous research on the blockchain has not focused extensively on a restricted number of peers in the blockchain network. Instead, it has mainly concentrated on predicting the reliability of blockchain systems.

In this work, we focus on collaborative filtering. Our inspiration comes from the collaborative filtering algorithm under the QoS prediction task. Regarding the collaborative filtering model on QoS prediction, we categorized them into three major methods. (1) Memory-based methods, which calculate the similarity between users and services using Pearson Correlation Coefficient (PCC) and predict missing values accordingly. Examples of these methods include UPCC [10], IPCC [11], and UIPCC [12], which combines both UPCC and IPCC approaches for better prediction. (2) Model-based methods, which learn the latent factors of users and services to predict missing values. Examples of these methods include Matrix Factorization (MF) [13] and Probabilistic Matrix Factorization (PMF) [14]. Classical model-based approaches like LN_LFM [15], CloudPred [16], and AMF [17] incorporate additional side information. Moreover, the Context-Sensitive Matrix Factorization approach (CSMF) [18] fully utilizes implicit and explicit contextual factors. (3) Hybrid approaches, which combine the memory-based and model-based CF approaches to achieve better prediction. These approaches are generally effective at estimating the overall structure that relates simultaneously to most or all services and also capture the totality of weak signals encompassed in all of a user’s QoS values. Recent studies combine neural networks with matrix completion, taking advantage of their non-linearity fitting ability to achieve more substantial collaborative information utilization. For instance, He et al. [19] combined a neural network and matrix factorization to predict the missing value. Wu et al. [20] proposed a Deep Neural Model (DNM), leveraging the rich contextual features for multiple-attribute QoS prediction. Moreover, most previous studies have not achieved good performance in blockchain systems under the sparse scenario since they do not fully utilize structural features (e.g., graph structural features). Here, we exploit the structural information more to achieve higher accuracy than the traditional method in blockchain services.

We observed better state-of-the-art recovery performance achieved by graph neural networks than that achieved by the memory-based one. Wang et al. [21] introduced a novel recommendation framework termed Neural Graph Collaborative Filtering (NGCF) that integrates the collaborative signal and connectivity information to the node embedding for effective collaborative filtering. He et al. [22] further optimized the framework of the NGCF and proposed LightGCN. Xu et al. [23] proposed ISPA-GNN, which leverages a novel graph-based collaborative filtering method with a subgraph sampling strategy. GNN-based models are yelling the powerful performance for CF-related tasks.

The graph attention mechanism [24] aims to help better capture graph structure features and has been shown to be a powerful tool for matrix decomposition and capture of collaborative information from graph data information. Its main idea is that by learning the relationship between nodes, different attention weights are assigned. This facilitates the efficient aggregation of information from neighboring nodes and the weighted combination of important neighboring node features with the representation of the target node, aiming to achieve better performance results. Despite great success in traffic flow [25], recommender system [26,27], and social network analysis [28,29] domains, we have no idea of the graph attention mechanism proposed for matrix completion. In traffic flow forecasting tasks, the authors of [30] showed that the attention mechanism can lead to better performance compared with that achieved by combing the graph neural networks and the recurrent neural networks model. We are now researching the attention mechanism in the neural matrix completion model.

The above work is summarized in Table 1. To the best of our knowledge, this is the first work to combine graph attention mechanisms and collaborative filtering to predict reliability in blockchain services. We were inspired by [31], which leverages the graph neural network and matrix factorization to estimate unknown reliability values in the data matrix. We see the opportunity and feasibility of the graph attention mechanism to incorporate graph-structured information. Therefore, we develop our GATCF method to achieve this.

## 3. Preliminary and Framework Overview

This section discusses preliminary data and provides an outlined overview of the framework with mathematical notation as shown in Table 2.

### 3.1. Problem Definition

**Definition: SuccessRate Value Calculation.** We regard blockchain services as blockchain peers composed of blockchain applications, while blockchain users are blockchain application developers who utilize blockchain services. To calculate the success rate for a request, we collect three QoS data points that serve as the foundation for the calculation. These data points are as follows:**Right Block Returns:** The peer should return the correct block hash that corresponds to the specified block height on the main blockchain. This ensures that the peer possesses accurate and up-to-date information.**Recent Block Height Returns:** The peer’s response should include a block height within an acceptable range of the highest block height in the batch. This criterion ensures the peer provides recent and relevant information. The parameter MaxBlockBack determines the maximum allowed difference between the block height returned by the peer and the highest block height in the batch. A value of 0 for MaxBlockBack indicates that the peer is considered reliable only if it returns the highest block in the batch.**Timely Response:** The round-trip time (RTT) of the request sent to the peer should be within an acceptable limit, denoted by MaxRTT. This criterion ensures that the peer responds in a timely manner, indicating its responsiveness and availability.

If a blockchain peer $P_{j}$ successfully responds to a batch request, it is recorded as a success in the $\;\mathrm{SuccessRequest}{\;}_{ij}$ category. Conversely, if the response is unsuccessful, it is recorded as a failure in the $\;\mathrm{FailureRequest}{\;}_{ij}$ category. The successful request rate of user *i* to peer *j* is calculated as follows:(1)$$y_{ij}=\frac{\;\mathrm{SuccessRequest}{\;}_{ij}}{\;\mathrm{SuccessRequest}{\;}_{ij}+\;\mathrm{FailureRequest}{\;}_{ij}},$$
where $y_{ij}$ denotes the SuccessRate of user *i* to peer *j*. By calculating the success rate, we can determine the effectiveness of the interaction between requester $u_{i}$ and peer $P_{j}$. By evaluating a peer’s performance against these criteria, the reliability of the peer can be assessed, and it can be determined whether the peer is suitable for participating in the blockchain network. These criteria help maintain the integrity and consistency of the blockchain by ensuring reliable and up-to-date information exchange among peers.

**Definition: SuccessRate Prediction.** To predict SuccessRate values between users and peers, a matrix is used where rows correspond to users and columns correspond to peers. However, since the matrix is sparse, we use a set $\Omega \;=\;\{\left(i,j\right)|Y_{ij}\neq 0\}$ to indicate the SuccessRate values between users and peers. Matrix completion is commonly used to infer the missing values by solving an optimization problem that minimizes the difference between observed and predicted values. In previous studies, given a sparse matrix Y, matrix completion has been used to infer the missing values by solving the following optimization problem:(2)$$\begin{array}{c} \Theta \;=\;\underset{\Theta }{argmin}\sum_{(i,j)\in \Omega }{\left(y_{ij}-f\left(u_{i},p_{j}|\Theta \right)\right)}^{2}, \end{array}$$
where Θ represents all the trainable weights in the matrix completion model, $u_{i}$ refers to the representation (i.e., embedding) of user *i*, while $p_{j}$ refers to the representation of peer *j*. These embeddings capture latent features that may be relevant to predicting missing values. *f* represents the interaction function of the model. The goal of recovering the sparsely observed data is to use the objective of matrix completion by focusing on the two-factor embeddings.

### 3.2. Overview

#### 3.2.1. Framework Overview

As illustrated in Figure 1, users send requests to their respective peers, and blockchain services (peers) respond by providing feedback on the quality of service (QoS) data. The feedback data comprises seven items, including the peer’s IP address, the user’s IP address, request time cost, response time cost, bulk request time cost, block height, and block hash. In the prediction server, the received feedback data is used to calculate the success rate based on the submitted information. These calculated success rates are then utilized to construct a user–service matrix representing the success rates for each user–service pair. Due to the limitation that users cannot request all services, the resulting matrix is highly sparse with numerous unknown values. To predict these unknown values, the GATCF model leverages the input context information. By employing GATCF, the unknown success rates can be estimated based on the known values. Once GATCF is applied, the request success rates for all users and services can be obtained. Consequently, the reliability of each blockchain service can be calculated using the service selector, which employs the predicted success rates to make informed decisions.

#### 3.2.2. Solution Overview

After introducing the whole process, we are now focused on the training process. Specifically, the training process in a stage has three main steps:

**Step 1: Embedding.** We initialize factor matrices for users and peers and retrieve their corresponding embeddings based on input indices. The embeddings capture latent features and characteristics of users and peers, providing a representation that allows making predictions or recommendations.

**Step 2: Transfer.** We construct bipartite graphs to capture complex relationships and interdependencies among users, peers, and contextual features. By utilizing graph convolution and message passing, we aggregate information from neighboring nodes. Attention weights are calculated to determine the importance of nodes, enabling us to update node embeddings. This process allows the transfer of learned graph structural features and capture of important neighborhood signals.

**Step 3: Interaction.** We concatenate the user and peer embeddings to create an input embedding matrix. We design a neural network-based prediction layer that models the interaction between user and peer embeddings. This prediction layer uses a neural network to capture complex relationships and produce the final score prediction value, allowing prediction of missing values or making recommendations based on user–peer interactions and contextual information.

GATCF achieves several advantages due to the graph attention mechanism and neural-network-based interaction function design. Specifically, (1) the graph attention approach involves an adaptive attention mechanism to model the relationship between nodes and adjusts the attention weights according to the similarity between users or peers. This enables the model to better capture the synergistic information between nodes and improve the performance of graph neural networks. (2) The use of the neural network offers the ability to learn arbitrary functions from data and possesses stronger nonlinear fitting capabilities. By employing multilayer perceptrons to learn user–peer interaction functions, neural network interaction functions can more accurately capture complex relationships between users and peers, resulting in improved performance.

## 4. Details of GATCF

Similar to most matrix completion approaches, our method primarily comprises embedding and interaction modules. Diverging from traditional approaches, we incorporate a transfer module. The embedding module (Section 4.1) takes in index values and retrieves the corresponding vectors from factor matrices. To incorporate graph patterns to achieve higher accuracy, the core idea behind this is to transfer the learned graph structural features via an attention mechanism. After the transformed embeddings are obtained, we input them to the interaction module and then output a prediction. We use neural networks to fully utilize the information encoded in embedding. The architecture is shown in Figure 2.

Recall that our goal is to extract collaborative information fully. Targeting the key components, we design a transfer module (Section 4.2) and the neural network interaction module (Section 4.3).

### 4.1. Embedding Module

Following the traditional collaborative filtering method, we let *R* denote the embedding dimension. We first randomly initialized factor matrices (a.k.a embedding matrices) $U\in R^{I\times R}$, $P\in R^{J\times R}$, for users and peers, respectively. Concretely, given indices (i,j) as the input, we can obtain the corresponding user *i* embedding and peer *j* embedding; $u_{i}$ is the i-th row of U, $p_{j}$ is the j-th row of P. These embeddings capture latent features or characteristics of users and peers, allowing the making of predictions or recommendations.

### 4.2. Transfer Module

In the context of reliability prediction on the blockchain, traditional methods often overlook the interconnectedness of different contextual features. For instance, an autonomous system may extend across multiple geographic regions or countries [7], while certain peer providers exclusively cater to specific autonomous systems or operate within specific geographic regions [31]. These intricate relationships can be accurately represented and modeled using graph structures. To capture the intricate connections between users/peers and their associated contexts, we employ a bipartite graph structure. By leveraging the edges connecting users/peers and contexts, nodes within the graph can access their neighboring nodes, allowing the extraction of important neighborhood signals.

In order to enhance the node features, it is crucial to convert the input embedding into higher-level representations. This transformation process requires a reliable and trainable linear transformation. To accomplish this, a shared linear transformation is employed, which is parameterized by an embedding matrix denoted as $W\in R^{F^{\prime }\times F}$. This matrix is applied to each node individually. Subsequently, self-attention is performed on the nodes using a shared attentional mechanism represented by $a:R^{F^{\prime }}\times R^{F^{\prime }}\to R$, which computes attention coefficients.
(3)$$\begin{array}{c} e_{ij}=a\left(W{\vec{h}}_{i},W{\vec{h}}_{j}\right). \end{array}$$

We incorporate the significance of node $j^{\prime }$s features to node *i* through attention mechanisms. In its general formulation, the model enables each node to attend to every other node without considering the structural information. However, we introduce graph structure by employing masked attention, which restricts the computation of the attention weights ($e_{ij}$) to nodes $j\in N_{i}$, where $N_{i}$ represents the neighborhood of node *i* in the graph. In our model, we specifically consider the first-order neighbors of node *i*, including the node itself. To make coefficients easily comparable across different nodes, we employ the softmax function to normalize them.
(4)$$\begin{array}{c} a_{ij}={\mathrm{softmax}}_{j}\left(e_{ij}\right)=\frac{\exp\left(e_{ij}\right)}{\sum_{k\in N_{i}}\exp\left(e_{ik}\right)}. \end{array}$$

Based on preliminaries, we first construct undirected graphs consisting of user nodes, user context nodes, peer nodes, and peer context nodes. Users belonging to the same context are connected to a specific context node. In the graphs denoted as $G_{u}$ and $G_{p}$, the first-order neighbor nodes of user/peer nodes represent their contextual features, while the second-order neighbor nodes refer to user/peer nodes with the same contextual features.

Secondly, we calculate the attention weights of each pair of nodes, user–user and peer–peer, by calculating the attention weights. The algorithm for calculating the attention weights between the nodes can be formulated as
(5)$$\begin{array}{c} a_{ij}=\frac{\exp\left(\mathrm{Leaky}\mathrm{ReLU}\left({\tilde{a}}^{T}\left[W_{u}u_{i}∥W_{u}u_{j}\right]\right)\right)}{\sum_{k\in N_{i}}\exp\left(\mathrm{LeakyReLU}\left({\tilde{a}}^{T}\left[W_{u}u_{i}∥W_{u}u_{k}\right]\right)\right)}, \end{array}$$
(6)$$\begin{array}{c} b_{ij}=\frac{\exp\left(\mathrm{LeakyReLU}\left({\tilde{b}}^{⊤}\left[W_{p}p_{i}∥W_{p}p_{j}\right]\right)\right)}{\sum_{k\in N_{i}}\exp\left(\mathrm{LeakyReLU}\left({\tilde{b}}^{⊤}\left[W_{p}p_{i}∥W_{p}p_{k}\right]\right)\right)}, \end{array}$$
where $a_{ij}$ represents the similarity between two users and $b_{ij}$ represents the similarity between two peers. $N_{i}$ denotes the set of neighboring nodes connected to a given node *i*, $\tilde{a}$ represents the initial feature vector of the user node, $\tilde{b}$ represents the initial feature vector of the peer node, $u_{i}$ represents the user’s embedding vector, and $p_{i}$ represents the peer’s embedding vector. $W_{u}$ and $W_{p}$ are the trainable weight of linear transformation in the transfer module. The ‖ symbol denotes the concatenation operation and ${}^{T}$ represents transposition. LeakyReLU is represented by the following equation:(7)$$\begin{array}{c} \mathrm{LeakyReLU}\;=\;\max(0,x)+\;\mathrm{leak}\;*\min(0,x), \end{array}$$
where leak is the hyperparameter that controls the slope of the negative part, which is a constant, less than 1.0, and is usually set to a smaller positive number.

After attention weight is calculated, we weigh and sum the feature vector of each node with the feature vectors of its neighboring nodes to obtain the new feature vector of that node. Formally, the update formula for the feature vector can be written as
(8)$$\begin{array}{c} u_{i}^{\prime }=\sigma \left(\sum_{j\in N_{i}}a_{ij}Wu_{j}\right), \end{array}$$
(9)$$\begin{array}{c} p_{j}^{\prime }=\sigma \left(\sum_{i\in N_{j}}b_{ij}Wp_{i}\right), \end{array}$$
where σ is the LeakyReLU activation function. $u_{i}^{\prime }$ and $p_{j}^{\prime }$ are the transferred embedding.

To help the model better focus on the important part of the input user embedding and peer embedding, similar to [32], we now extend our mechanism to employ multi-head attention. Fundamentally, k-independent attention mechanisms execute the transformation depicted in Equations (8) and (9). Subsequently, their outputs are amalgamated through concatenation, yielding the ultimate output feature representation.
(10)$$\begin{array}{c} u_{i}^{\prime }{=∥}_{k=1}^{K}\sigma \left(\sum_{j\in N_{i}}a_{ij}^{k}W_{u}^{k}u_{j}\right), \end{array}$$
(11)$$\begin{array}{c} p_{j}^{\prime }{=∥}_{k=1}^{K}\sigma \left(\sum_{i\in N_{j}}b_{ij}^{k}W_{p}^{k}p_{i}\right), \end{array}$$
where $a_{ij}^{k}$ and $b_{ij}^{k}$ are normalized attention coefficients computed by kth user attention mechanism($a^{k}$) and kth peer attention mechanism ($b^{k}$). σ is the LeakyReLU activation function. $W_{u}^{k}$ and $W_{p}^{k}$ are the corresponding trainable weight of the input linear transformation. We suppose that the final outputs $u_{i}^{\prime }$ and $p_{j}^{\prime }$ consist of K×R features for each node.

In particular, when using multi-head attention on the final layer of the network for prediction, concatenation is not a viable option. Instead, we use an averaging operation:(12)$$\begin{array}{c} u_{i}^{\prime }=\sigma \left(\frac{1}{K}\sum_{k=1}^{K}\sum_{j\in N_{i}}a_{ij}^{k}W_{u}^{k}u_{j}\right), \end{array}$$(13)$$\begin{array}{c} p_{j}^{\prime }=\sigma \left(\frac{1}{K}\sum_{k=1}^{K}\sum_{i\in N_{j}}b_{ij}^{k}W_{p}^{k}p_{i}\right), \end{array}$$
where *k* is the number of heads in the transfer module.

### 4.3. Interaction Module

In our matrix completion, we turn our attention to modeling the relationship representing the interaction between the user and their peer’s embedding vectors. We can directly employ the inner-product-based interaction function to predict the missing value, formulated as
(14)$$\begin{array}{c} \hat{y_{ij}}\;=\;\sigma \left(\sum_{r=1}^{R}u_{ir}p_{jr}\right), \end{array}$$
where σ(x)=1/(1+exp(−x)) is a non-linear activation function, mostly used to enable the model to capture non-linearity interaction patterns. This method forces the module to focus on embeddings without interfering with the interaction function. However, there are many other kinds of more expressive interaction forms. For example, Cheng et al. [33] proposed that cross-product features between historical user behaviors and candidate items are widely used in click-through rate prediction. To obtain higher accuracy, complex relationships need to be modeled. It could not be more appropriate to employ a neural-network-based interaction function. The deep interest network [34] highlights that user interests vary and an attention-based network structure can produce different user vectors for different candidate items. Therefore, in order to improve the accuracy of reliability prediction for nearby users and peers, we leverage the neural network interaction function instead of the inner product interaction function. Concretely, we first concatenate the user embeddings and peer embeddings together:(15)$$\begin{array}{c} H\;=\;U^{\prime }\;\|\;P^{\prime }, \end{array}$$
where $U^{\prime }$ and $P^{\prime }$ are the transferred embedding of the user and peer. H is the input embedding of the prediction layer which is introduced further. ‖ is the concatenation operation in this process. After obtaining the input embeddings, the neural-network-based prediction layer is constructed and shown in Figure 3. Specifically, the prediction layer is expressed in the following formula:(16)$$\begin{array}{c} \hat{y}\;=\;g\left(H|\Theta_{g}\right), \end{array}$$
where g is the prediction layer of the model, and $\Theta_{g}$ is the trainable parameters of the prediction layer. $\hat{y}$ represents the final prediction value.

### 4.4. Model Training

With all modules introduced, we now consider how to train our model to complete the entire matrix. Specifically, the matrix completion process minimizes loss L as
(17)$$L=\sum_{i,j\in \Sigma }\left(y_{ij}-f\left(i,j|\Theta \right)\right);$$
this is a mean absolute error between the predictions and the labels. Θ is the trainable parameters of the model. $y_{ij}$ represents the SuccessRate of user *i* to peer *j*. *f* denotes the entire modules. By minimizing this loss function using optimization algorithms, the GATCF model can learn to complete the missing entries in the matrix accurately.

### 4.5. Complete Solution

The execution process of GATCF can be described as follows. The GATCF (Graph Attention collaborative filtering) model is a complete solution that combines matrix factorization, graph attention mechanism, and neural networks. It captures latent features of users and peers through the Embedding Module, incorporating graph patterns and neighborhood information via the transfer module. After obtaining ultimate embeddings, we can predict the missing value by the interaction function. The interaction module models the interaction between user and peer embeddings using a neural network-based function. This comprehensive approach enhances collaborative filtering and improves reliability prediction, making the GATCF model a powerful solution for various recommendation and prediction tasks. To maintain optimal model performance, we initiate both the training and estimation processes in response to retraining conditions.

## 5. Experiments

In this section, the authors describe the experiment settings and objectives for evaluating the performance of their proposed GATCF framework. They aim to answer the following research questions:**RQ1** How well does GATCF perform in the real world?**RQ2** How do different components affect GATCF?**RQ3** How do interaction modules affect GATCF?**RQ4** How do hyper-parameters affect GATCF?

### 5.1. Experiment Settings

#### 5.1.1. Datasets

In our study, we utilized a real-world dataset originally proposed in [7]. This comprehensive dataset encompasses 100 blockchain users and 200 blockchain peers, collected using a framework [35] that includes a data collector and log parser. This framework allows developers convenient storage of data in their own database instead of on the blockchain. The blockchain peers in this dataset span 21 countries, while the users hail from 15 different nations. With over 2 million test cases, this dataset provides a rich source of information for our analysis.

#### 5.1.2. Baselines

We compare our model with three types of methods, including two classical memory-based collaborative filtering methods and two hybrid-based collaborative filtering methods:UMEAN [7]: This is a user-based approach that employs the average success rate of the current requester on other blockchain peers for the prediction.IMEAN [7]: This is an item-based approach that employs the average success rate of the blockchain peers observed by other requesters for the prediction.UPCC [10]: This is a user-based collaborative filtering approach that employs PCC (Pearson Correlation Coefficient) to calculate similarities between users, then predicts missing values by considering the similar user and its neighbors.IPCC [11]: This is a memory-based method that is similar to UPCC and also uses pcc to calculate similarities between peers and identifies the peers’ neighbors to predict the missing value.UIPCC [12]: This is a memory-based method that combines the mechanics of UPCC and IPCC to predict missing values.PMF [14]: This is one of the basic algorithms for recommender systems that handle very sparse and unbalanced data sets well by complete treatment of Bayes’ theorem.MF [13]: This is a basic matrix factorization model that expresses the connection between the user and the item by using the inner product to interact with the user-specific radial quantity with the item factor, without utilizing context or bayes’ theorem.MNCF [36]: This is a model that combines a neural network with matrix factorization to perform collaborative filtering for the latent feature vectors of users and introduces multi-task learning for sharing different parameters.FL-MFGM [37]: This is a privacy-preserving and high-accuracy blockchain reliability prediction model that protects user privacy by uploading the gradients of matrix factorization based on federated learning architecture.GraphMF [31]: This is a graph neural network-based model which combines GNNs and collaborative filtering to extract features to estimate missing QoS values in the data matrix.

#### 5.1.3. Metrics

Let $y_{ij}$ and ${\hat{y}}_{ij}$ denote the ground-truth value and the estimated value. *N* denotes the number of testing samples. Ω denotes the indices of unobserved values. To evaluate the recovery performance on unobserved values, two commonly used metrics are applied:**Mean Absolute Error. MAE** is calculated by
(18)$$\begin{array}{c} \mathrm{MAE}\;=\;\frac{1}{N}\sum_{\left(i.j\right)\in \bar{\Omega }}\left|y_{ij}-{\hat{y}}_{ij}\right|. \end{array}$$**Root Mean Square Error. RMSE** takes the form
(19)$$\begin{array}{c} \mathrm{RMSE}\;=\;\frac{\sum_{\left(i.j\right)\in \bar{\Omega }}\left|y_{ij}-{\hat{y}}_{ij}\right|}{\sum_{\left(i.j\right)\in \bar{\Omega }}\left|y_{ij}\right|}. \end{array}$$

For both metrics, **MAE** and **RMSE**, smaller values indicate better prediction performance.

### 5.2. Implementation Details

The GATCF https://github.com/ZengYuXiang7/GATCF (accessed on 8 May 2023) is implemented in PyTorch and performance evaluation on a workstation equipped with a 2.30 GHz Intel Core i7-11800H CPU (Santa Clara, CA, USA) and 32 GB of memory, as well as an NVIDIA GeForce RTX 3080 laptop GPU (Santa Clara, CA, USA) with 16 GB of memory, running on Ubuntu 18.04. To prevent overfitting, the maximum number of epochs was set to 100, and early stopping was employed with a patience value of 20. The batch size for the model was set to 4096 and the AdamW [38] optimizer was used to optimize all models. We tested our proposed model 20 times in every density and took the average of the twenty.

### 5.3. Data Preprocessing

We use MaxRTT to represent the maximum tolerable request round-trip time and MaxBlockBack to represent the maximum tolerable return block backward relative to the latest block. By adjusting the parameters of MaxBlockBack and MaxRTT, we can simulate real-world scenarios in our blockchain system. For example, when evaluating accuracy in situations with stringent requirements for confirming blockchain data, we set four different combinations of MaxBlockBack and MaxRTT for our experiments: (MaxBlockBack = 0, MaxRTT = 1000), (MaxBlockBack = 12, MaxRTT = 1000), (MaxBlockBack = 12, MaxRTT = 2000), and (MaxBlockBack = 100, MaxRTT = 5000). These settings allow accurate assessment of the performance of our system under a range of conditions, and details are shown in Table 3.

Regarding the datasets mentioned above, we divide them into two disjoint sets: (1) the training set, used to train our model and baselines, and (2) the test set, applied to evaluate model performance and report the experiment results.

### 5.4. Performance Comparison (RQ1)

Table 4 compares the performance of the GATCF model with baseline approaches across different datasets, and several key observations can be made. The GATCF model outperforms the baseline methods by achieving lower **MAE** and **RMSE** values across four different datasets and evaluation levels, indicating its superiority in accurately predicting user–peer reliability with reduced errors. Firstly, in the “MaxBlockBack = 0, MaxRTT = 1000” setting, the GATCF model consistently achieves the lowest **MAE** and **RMSE** values across all evaluation levels, indicating its superior predictive accuracy and minimized errors. This signifies that the GATCF model is highly effective in capturing and modeling complex relationships between users and peers in scenarios with minimal block backlogs and round-trip times. Similarly, in the “MaxBlockBack = 12, MaxRTT = 1000” and “MaxBlockBack = 12, MaxRTT = 2000” settings, the GATCF model outperforms the baseline methods, displaying significantly lower **MAE** and **RMSE** values. This highlights the model’s capability to accurately predict user–peer relationships when there is a moderate block backlog and round-trip time. Furthermore, even in the challenging scenario of “MaxBlockBack = 100, MaxRTT = 5000,” where the block backlog and round-trip time are relatively high, the GATCF model consistently demonstrates the lowest **MAE** and **RMSE** values across all evaluation levels. This indicates the model’s robustness and effectiveness in capturing the intricate dynamics of user–peer relationships in such demanding scenarios. Specifically, GATCF improves **MAE** over the best of baselines by 10.14%, 26.97%, 36.89%, and 75.39% on four datasets and improves **RMSE** by 9.94%, 20.61%, 28.75% and 53.11% on four datasets. This proves that GATCF is very effective in learning complex and non-linear graph patterns under a low matrix density.

Through the conducted experiment, we determine a key feature of GATCF— effectiveness. Regarding the other models, they are all weaker than GATCF in the sparse case, although they are all able to achieve excellent performance at high sampling densities according to studies conducted to examine them. The exceptional performance of the GATCF model can be attributed to its innovative graph attention mechanism (GAT) architecture, which leverages multiple attention heads to capture diverse dependencies and interactions between users and peers. By effectively attending to relevant features and relationships, the GATCF model can make highly accurate predictions.

### 5.5. Ablation Study (RQ2)

To show the effectiveness of our attention-based transfer module in extracting more collaborative information, we compare the GATCF with its variant GATCF-m, which only uses a neural-network-based interaction module. Figure 4 shows the recovery performance of our GATCF (w/attention) with its variant GATCF-m (w/o attention). In four scenarios, GATCF’s performance consistently surpasses its variant without the attention transfer module under different sampling ratios. Specifically, regardless of the specific variations in “MaxBlockBack” and “MaxRTT” parameters, incorporating an attention mechanism leads to lower **MAE** and **RMSE** values, indicating the model with attention consistently exhibits superior predictive capabilities with lower error values.

### 5.6. Impact of the Interaction Module (RQ3)

In this section, we focus on the impact of the interaction function module. The interaction function can capture some potential factors between user embeddings and peer embeddings. We investigate its influence on prediction accuracy via inner-product- or neural-network-based interaction modules.

We also show other results in Figure 5. We learned that the performance of GATCF with a neural-network-based interaction function achieves higher accuracy than GATCF with an inner-product-based interaction function. Whether it is in the “MaxBlockBack = 0, MaxRTT = 1000,” “MaxBlockBack = 12, MaxRTT = 1000,” “MaxBlockBack = 12, MaxRTT = 2000,” or “MaxBlockBack = 100, MaxRTT = 5000” settings, the neural network interaction method demonstrates lower **MAE** values, indicating higher accuracy. This highlights the superiority of the neural network approach in capturing complex relationships between users and peers. By effectively capturing and learning intricate patterns and dependencies in user–peer data, the neural network method enables more accurate predictions and reduces prediction errors. Moreover, we also observed some instability in model training based on the interaction function of the inner product. Based on our experiments, we reinforced our choice of neural network interaction functions.

### 5.7. Impact of the Hyper-Parameters (RQ4)

**Impact of Dimension.** Feature dimension represents how many hidden factors affect the prediction values. It has a crucial impact on both the requirement for computational resources and model accuracy. We fixed other hyperparameters, set the sampling ratio to 5% on all datasets, and drew the recovery performance curve. In Figure 6, a key observation can be made. Firstly, in the “MaxBlockBack = 0, MaxRTT = 1000” setting, both **MAE** and **RMSE** values show a decreasing trend as the dimensionality increases from 16 to 256. This indicates that increasing the dimensionality leads to improved model accuracy and reduced prediction errors. Similarly, in the “MaxBlockBack=12, MaxRTT = 1000”, “MaxBlockBack = 12, MaxRTT = 2000” and “MaxBlockBack = 100, MaxRTT = 5000” settings, both **MAE** and **RMSE** values exhibit a decreasing trend as the dimensionality increases. The results suggest that increasing the dimensionality has a positive impact on the performance of the model in terms of **MAE** and **RMSE**. Higher dimensionalities enable the model to capture more complex patterns and dependencies in user–eer relationships, resulting in improved accuracy and reduced prediction errors, highlighting the importance of considering higher dimensionalities in modeling user–peer relationships. Therefore, we set the dimension to 256 for four different configuration datasets.

**Impact of Attention Head.** The number of heads indicates the number of heads included in the multi-headed attention mechanism. It is an important hyperparameter in our model. It determines memory consumption and affects training time and model accuracy. In Figure 7, the results are as follows: for the “MaxBlockBack = 0, MaxRTT = 1000” setting, the **MAE** and **RMSE** values exhibit fluctuations without a clear trend as the attention heads increase. This suggests that the choice of attention head count has a limited impact on the model’s performance in this specific scenario. However, in the “MaxBlockBack = 12, MaxRTT = 1000”, “MaxBlockBack = 12, MaxRTT = 2000” and “MaxBlockBack = 100, MaxRTT = 5000” settings, increasing the number of attention heads leads to higher **MAE** and **RMSE** values, indicating a potential degradation in model performance. One possible reason for the decrease in model performance when the number of attention heads increases is excessive attentional distraction. Therefore, it is advisable to use smaller attention head counts in these cases to maintain better model accuracy. Notably, the model achieves better performance when the attention head count is set to two for the “MaxBlockBack = 0, MaxRTT = 1000,” “MaxBlockBack = 12, MaxRTT = 1000,” and “MaxBlockBack = 12, MaxRTT = 2000” settings. In light of these findings, we recommend using an attention head count of two for all datasets, as it consistently yields improved performance across multiple settings. This decision ensures an optimal trade-off between accuracy and computational efficiency, enhancing the overall effectiveness of the model.

## 6. Conclusions and Future Work

### 6.1. Conclusions

In the BaaS environment, it is critical to select reliable blockchain services for developing blockchain-based applications. To achieve this, we propose a novel Graph Attention Matrix Factorization (GATCF) method that achieves higher accurate prediction performance in blockchain service reliability prediction. To achieve this, we first feed the users and peers embedding to the embedding transfer module to extract more graph structural features. Then, we leverage the interaction module to model the interaction relationship between the user’s latent factor vector and the peer’s latent factor vector. Large-scale experiments show that our GATCF outperforms all of the baseline models.

### 6.2. Future Work

For future work, we plan to extend our framework to handle more scenarios such as dynamic data and heterogeneous data sources. (1) Dynamic data usually refers to data that are constantly changing, such as real-time transaction information, real-time transaction volume in a blockchain network, etc. These data need to be updated and processed in real-time in order to reflect the latest information in a timely manner. (2) For heterogeneous data, blockchain applications are becoming more and more common in enterprise business operations, but most of the data in enterprise business activities come from different data sources with inconsistent information representations and serious ambiguities between the same blockchain, making it difficult to assess the consistency, trustworthiness, and value of the information.

In our work, once we have thoroughly explored the performance of our model, we will investigate other alternative methods that can achieve even higher levels of accuracy.

## Figures and Tables

**Figure 1 sensors-23-06775-f001:**
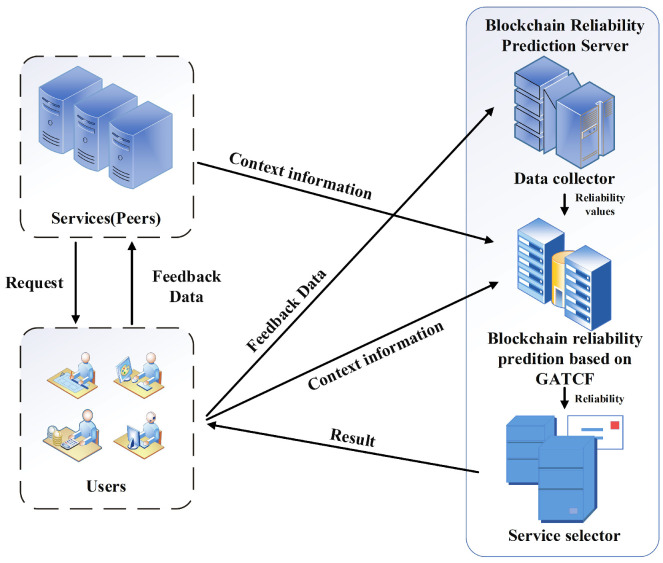
Reliability Prediction Framework for Blockchain Services.

**Figure 2 sensors-23-06775-f002:**
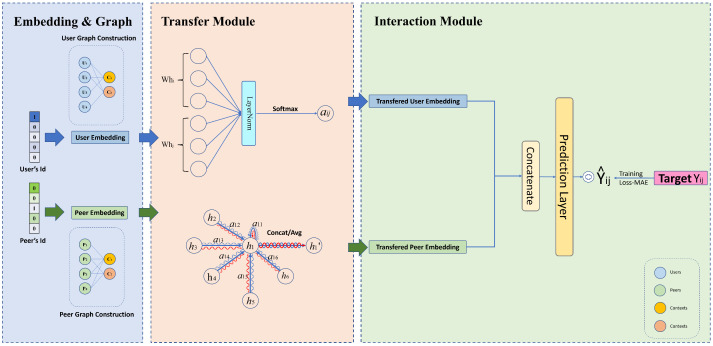
Illustration of the proposed GATCF model.

**Figure 3 sensors-23-06775-f003:**
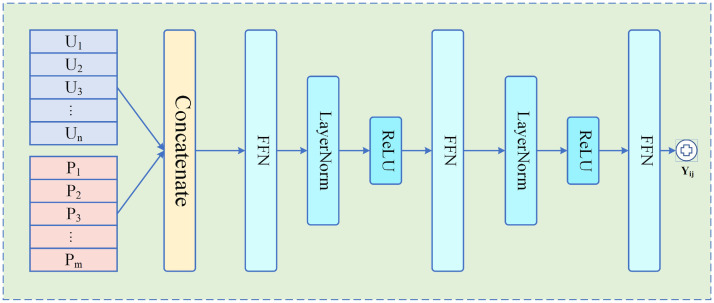
The Prediction Layer.

**Figure 4 sensors-23-06775-f004:**
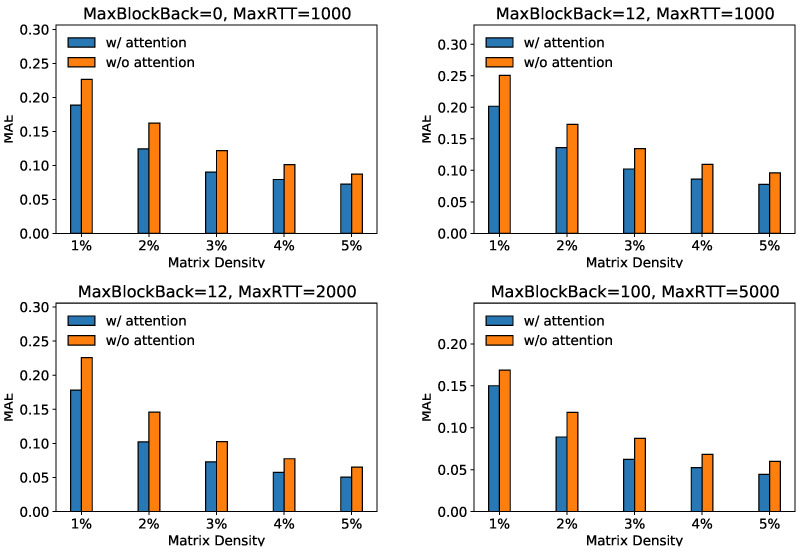
Ablation study on Attention Modules.

**Figure 5 sensors-23-06775-f005:**
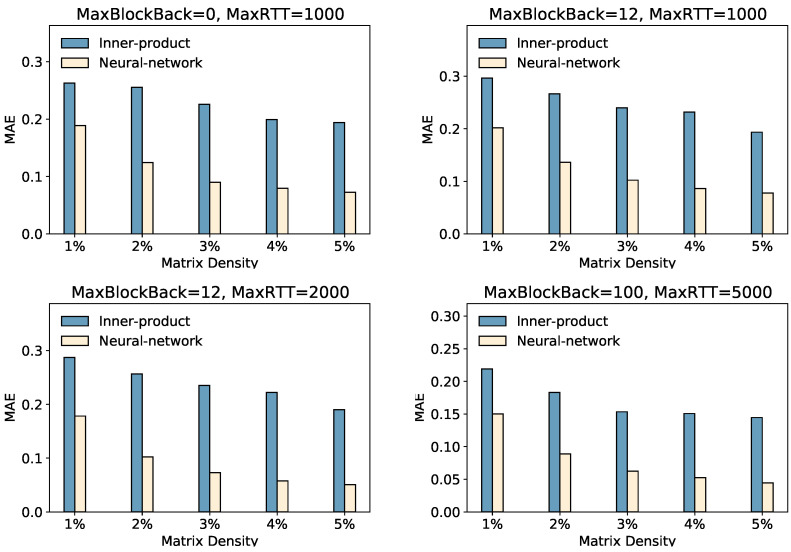
Performance of Error with Different Interaction Module.

**Figure 6 sensors-23-06775-f006:**
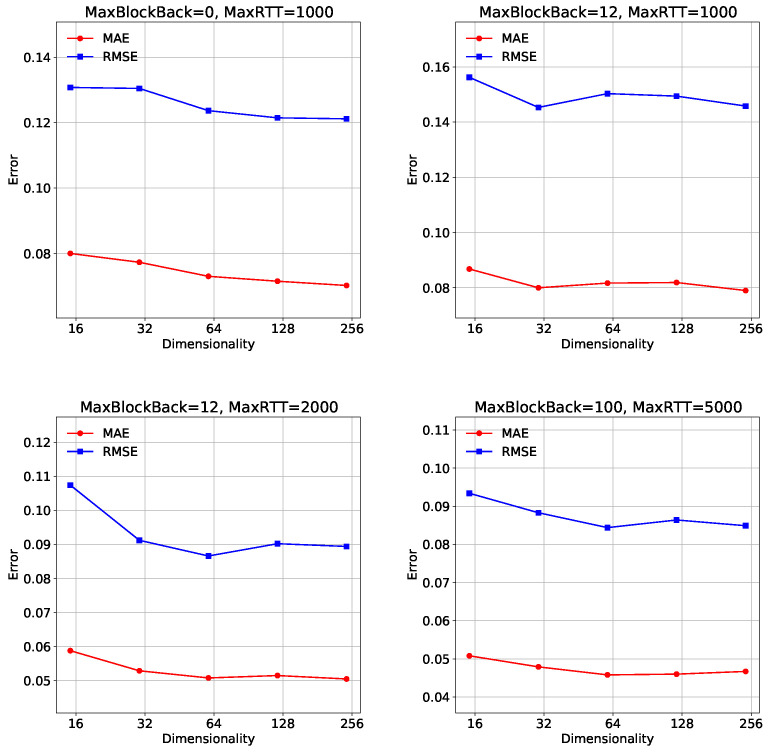
Analysis of Dimension.

**Figure 7 sensors-23-06775-f007:**
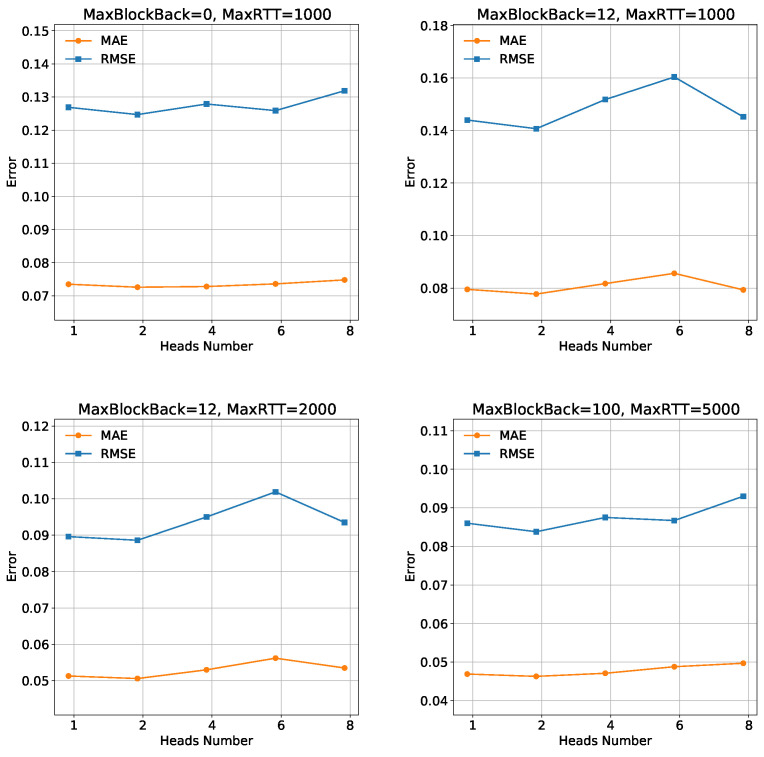
Analysis of Head.

**Table 1 sensors-23-06775-t001:** Exploring the Advantages and Limitations of Previous Methods in Predictive Modeling.

Method	Advantages	Limitations
Memory-based	- Simple and easy to implement. - Calculates similarity between users and peers. - Effectively utilizes local information.	- Computation time for similarity in large-scale systems. - Lower prediction performance for sparse datasets. - Suffers from cold-start and scalability problems.
Model-based	- Ability to utilize global information. - Capture potential features of users and peers.	- Need for large amounts of training data. - High computational complexity, and difficulty in handling dynamically changing reliability data.
Hybrid approaches	- Combination of memory-based and model-based methods. - Estimation of the overall structure.	- Increases the complexity of the model. - Requires tuning of more parameters.
Graph neural networks	- Better capture of graph structure features. - Advanced performance in collaborative filtering.	- Requirement of significant computational resources and training data. - Lower prediction performance for sparse datasets.
Attention mechanism	- Effective capture of node relationships and importance. - Excellent performance in multiple domains.	- Requirement of significant computational resources and training data.

**Table 2 sensors-23-06775-t002:** Selected Notations and Definitions.

Notation	Definition
$y_{ij}$	The SuccessRate of user *i* to peer *j*.
Θ	All the trainable weights in the matrix completion model.
$u_{i}$	The learned representation (i.e., embedding) of user *i*.
$p_{j}$	The learned representation of service *j*.
$a_{ij}$	The similarity between two peers.
$b_{ij}$	The interaction function of different models.
$N_{i}$	The set of neighboring nodes connected to a given node *i*.
$\tilde{a}$	The initial feature vector of the user node.
$\tilde{b}$	The initial feature vector of the service node.
$W_{u}$	The user’s embedding vector.
$W_{p}$	The peer’s embedding vector.
σ	The LeakyReLU activation function.
$u_{i}^{\prime }$, $p_{j}^{\prime }$	The transferred embedding.
*g*	The prediction layer of the model.
$\Theta_{g}$	The trainable parameters of the prediction layer.

**Table 3 sensors-23-06775-t003:** Different settings of MaxBlockBack and MaxRTT.

Dataset Setting	1	2	3	4
MaxBlockBack	0	12	12	100
MaxRTT	1000	1000	2000	5000

**Table 4 sensors-23-06775-t004:** Comparison of **MAE** and **RMSE** among GATCF and baseline approaches.

Datasets	Model	MAE					RMSE				
		1%	2%	3%	4%	5%	1%	2%	3%	4%	5%
0–1000	UMEAN	0.3782	0.3624	0.3423	0.3349	0.3282	0.4713	0.4421	0.4061	0.3909	0.3787
	IMEAN	0.2027	0.1368	0.1049	0.0799	0.0748	0.3398	0.2582	0.2012	0.1458	0.1320
	UPCC	0.3782	0.3621	0.3399	0.3257	0.3026	0.4713	0.4419	0.4044	0.3841	0.3591
	IPCC	0.2027	0.1368	0.1049	0.0794	0.0735	0.3398	0.2583	0.2013	0.1460	0.1325
	UPICC	0.1987	0.1392	0.1028	0.0874	0.0816	0.3261	0.2470	0.1800	0.1435	0.1312
	PMF	0.7098	0.7057	0.7022	0.6986	0.6954	0.7190	0.7167	0.7148	0.7129	0.7112
	MF	0.4155	0.2580	0.1807	0.1465	0.1210	0.5309	0.3794	0.2761	0.2242	0.1760
	MNCF	0.3894	0.1589	0.1188	0.0910	0.0755	0.4208	0.2565	0.2048	0.1651	0.1420
	MFGM	0.4145	0.3213	0.1875	0.1144	0.0854	0.5364	0.4447	0.2755	0.1694	0.1276
	GraphMF	0.2249	0.1514	0.1085	0.0910	0.0796	0.3026	0.2256	0.1687	0.1419	0.1278
	**GATCF**	**0.1886**	**0.1242**	**0.0901**	**0.0793**	**0.0726**	**0.2777**	**0.2052**	**0.1682**	**0.1332**	**0.1247**
12–1000	UMEAN	0.4297	0.4033	0.3781	0.3667	0.3604	0.5337	0.4936	0.4503	0.4270	0.4131
	IMEAN	0.2701	0.1728	0.1227	0.0974	0.0869	0.4333	0.3211	0.2377	0.1866	0.1612
	UPCC	0.4296	0.4031	0.3758	0.3585	0.3345	0.5337	0.4935	0.4487	0.4210	0.3938
	IPCC	0.2701	0.1728	0.1228	0.0977	0.0879	0.4333	0.3211	0.2377	0.1867	0.1615
	UPICC	0.2645	0.1748	0.1296	0.1067	0.0973	0.4138	0.3048	0.2276	0.1813	0.1587
	PMF	0.7633	0.7615	0.7570	0.7519	0.7438	0.7749	0.7739	0.7715	0.7687	0.7668
	MF	0.5496	0.3047	0.1997	0.1607	0.1455	0.6648	0.4630	0.2992	0.2367	0.2048
	MNCF	0.4107	0.1928	0.1389	0.1041	0.0850	0.4373	0.3124	0.2428	0.1922	0.1651
	MFGM	0.5380	0.4209	0.2209	0.1439	0.0918	0.6644	0.5448	0.3050	0.2041	0.1357
	GraphMF	0.2545	0.1930	0.1162	0.0982	0.0809	0.3493	0.2780	0.1914	0.1589	0.1403
	**GATCF**	**0.2017**	**0.1361**	**0.1022**	**0.0861**	**0.0777**	**0.3025**	**0.2305**	**0.1835**	**0.1567**	**0.1281**
12–2000	UMEAN	0.4455	0.4010	0.3765	0.3645	0.3613	0.5505	0.4916	0.4479	0.4245	0.4162
	IMEAN	0.2676	0.1399	0.0854	0.0624	0.0535	0.4442	0.2913	0.1949	0.1320	0.1069
	UPCC	0.4455	0.4006	0.3736	0.3541	0.3312	0.5505	0.4913	0.4458	0.4168	0.3937
	IPCC	0.2676	0.1399	0.0855	0.0626	0.0541	0.4442	0.2913	0.1949	0.1320	0.1072
	UPICC	0.2652	0.1455	0.0959	0.0752	0.0665	0.4246	0.2762	0.1871	0.1311	0.1089
	PMF	0.7878	0.7815	0.7763	0.7722	0.7671	0.7923	0.7889	0.7862	0.7840	0.7813
	MF	0.5878	0.3048	0.1832	0.1325	0.1164	0.6985	0.4501	0.2927	0.2050	0.1668
	MNCF	0.4134	0.1998	0.1351	0.0961	0.0754	0.4399	0.3204	0.2381	0.1856	0.1490
	MFGM	0.5796	0.4087	0.2130	0.1058	0.0653	0.6953	0.5267	0.2935	0.1474	0.0916
	GraphMF	0.2437	0.1531	0.0931	0.0699	0.0547	0.3357	0.2409	0.1507	0.1072	0.0897
	**GATCF**	**0.1782**	**0.1022**	**0.0730**	**0.0576**	**0.0506**	**0.2847**	**0.1871**	**0.1372**	**0.1037**	**0.0886**
100–5000	UMEAN	0.4130	0.3252	0.2749	0.2635	0.2555	0.5293	0.4317	0.3592	0.3382	0.3253
	IMEAN	0.3178	0.1561	0.0860	0.0686	0.0511	0.5001	0.3261	0.2037	0.1625	0.1114
	UPCC	0.4129	0.3249	0.2729	0.2572	0.2360	0.5293	0.4315	0.3576	0.3329	0.3089
	IPCC	0.3178	0.1561	0.0861	0.0589	0.0494	0.5001	0.3261	0.2037	0.1318	0.1065
	UPICC	0.3102	0.1568	0.0907	0.0643	0.0560	0.4724	0.3041	0.1912	0.1274	0.1049
	PMF	0.7991	0.7915	0.7904	0.7877	0.7832	0.8052	0.8012	0.8006	0.7992	0.7968
	MF	0.7093	0.3175	0.1895	0.1371	0.1172	0.7804	0.4609	0.3012	0.2000	0.1575
	MNCF	0.4041	0.2221	0.1573	0.1050	0.0788	0.4345	0.3446	0.2716	0.1990	0.1586
	MFGM	0.7039	0.4324	0.2033	0.0944	0.0636	0.7815	0.5255	0.2616	0.1259	0.0884
	GraphMF	0.1954	0.1568	0.0878	0.0715	0.0459	0.2935	0.2465	0.1521	0.1157	0.0780
	**GATCF**	**0.1501**	**0.0890**	**0.0624**	**0.0525**	**0.0444**	**0.2433**	**0.1610**	**0.1172**	**0.0982**	**0.0753**

## Data Availability

Not applicable.

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
