# Peer review of "GATCF: Graph Attention Collaborative Filtering for Reliable Blockchain Services Selection in BaaS"

_sensors, 2023, doi:10.3390/s23156775_

Round 1
Reviewer 1 Report
The paper introduces a new model that combines graph attention mechanisms with collaborative filtering and investigates the impact of different interaction functions on the overall performance of the model.
The authors argue in this paper that " GATCF can predict the reliability of a blockchain peer without invoking the blockchain peer, thus preventing the resource waste and economic loss caused by establishing a connection with an unreliable blockchain peer."
- The proposed framework outperforms most existing approaches in terms of predictive performance, showing that it outperforms existing reliability prediction models.
The paper argues that GATCF can be applied beyond blockchain technology to the field of service-oriented computing. The authors suggest that the proposed framework can be used to predict the reliability of services in a variety of domains, including the Internet of Things, edge computing, and web services. The authors also state that the proposed model can be extended to handle more scenarios, such as dynamic data and heterogeneous data sources, further increasing its applicability to various service-oriented computing domains. Overall, the paper suggests that GATCF could be a useful tool for selecting trusted services in a wide range of service-oriented computing domains beyond blockchain technology.
Reviewer 2 Report
Suggestions for Enrichment:
-
Conduct a comprehensive comparative analysis of the GATCF model with existing state-of-the-art methods in reliability prediction for blockchain services. This analysis should consider various performance metrics and datasets to highlight the strengths and weaknesses of the proposed approach.
-
Elaborate on the practical implementation challenges of the GATCF model, such as scalability, real-time processing, and integration with existing blockchain platforms. Discuss potential solutions and trade-offs to address these challenges and provide insights into the model's practical deployment.
-
Extend the discussion on handling dynamic and heterogeneous data sources in the proposed framework. Detail how the GATCF model can adapt to changing network conditions and accommodate different types of blockchain networks.
Suggested References to Enrich the Paper:
- https://journals.mesopotamian.press/index.php/CyberSecurity/article/view/68
- https://ieeexplore.ieee.org/abstract/document/7917130/
- https://mesopotamian.press/journals/index.php/CyberSecurity/article/view/90
Including these references will provide additional insights, background information, and related work to further enrich the discussion and strengthen the paper's overall contribution.
N/A
Reviewer 3 Report
A collaborative filtering-based matrix completion model, called Graph Attention Collaborative Filtering (GATCF) is proposed in this paper, which leverages both graph attention and collaborative filtering techniques to recover the missing values in the data matrix effectively. and the authors declared that by incorporating graph attention into the matrix completion process, GATCF can effectively capture the underlying dependencies and interactions between users or peers, and thus mitigate the data sparsity scenarios.
In my opinion, the paper's topic is interesting and the paper is well-organized, the authors need to revise the paper before possible acceptance. the comments of this reviewer are listed below:
1) literature review needs to be rich more, and authors need to consider the advantages and disadvantages of prior papers.
2) Authors can add a table for literature review and express the advantages, disadvantages, and limitations of previous work.
3) Add a flowchart to show the procedure of the suggested method.
4) Figure 4 is not viewed completely and some part of the figure is missing.
5) Add an abbreviations list.
6) References for equations 18 and 19 is missing, add references for these equations.
7) Quality of figure 6 and 7 are low, improve the quality of figures.
8) Express all details of the test system in a table.
9) All references are not cited in the paper, for example references 29, 32. Double check all references to be cited in the paper in order.
Dear Editor,
The English of paper is good.
Round 2
Reviewer 3 Report
The authors revised the paper well, and there are no more comments.
In general, the quality is good.